# Breakdown of the scaling relation of anomalous Hall effect in Kondo lattice ferromagnet USbTe

Hasan Siddiquee[1], Christopher Broyles[1], Erica Kotta[2], Shouzheng Liu[2], Shiyu Peng[3], Tai Kong[4], Byungkyun Kang[5], Qiang Zhu[5], Yongbin Lee[6], Liqin Ke[6], Hongming Weng[3], Jonathan D. Denlinger[7], L. Andrew Wray[2] & Sheng Ran[1] ✉

The interaction between strong correlation and Berry curvature is an open territory of in the field of quantum materials. Here we report large anomalous Hall conductivity in a Kondo lattice ferromagnet USbTe which is dominated by intrinsic Berry curvature at low temperatures. However, the Berry curvature induced anomalous Hall effect does not follow the scaling relation derived from Fermi liquid theory. The onset of the Berry curvature contribution coincides with the Kondo coherent temperature. Combined with ARPES measurement and DMFT calculations, this strongly indicates that Berry curvature is hosted by the flat bands induced by Kondo hybridization at the Fermi level. Our results demonstrate that the Kondo coherence of the flat bands has a dramatic influence on the low temperature physical properties associated with the Berry curvature, calling for new theories of scaling relations of anomalous Hall effect to account for the interaction between strong correlation and Berry curvature.

The anomalous Hall effect (AHE) can arise from two different mechanisms: extrinsic processes due to scattering effects, and an intrinsic mechanism connected to the Berry curvature associated with the Bloch waves of electrons. While large Berry curvature arises when the inversion or time-reversal symmetry of the material is broken, in the clean limit skew scattering can dominate and cause large AHE. To distinguish different mechanisms, scaling analysis has been developed and widely used in various systems[1–15]. Skew scattering depends on the scattering rate, leading to a quadratic dependence of anomalous Hall conductivity on the longitudinal conductivity, $\sigma_{xy}^{a} \sim \sigma_{xx}^{2}$, while $\sigma_{xy}^{a}$ originating from the intrinsic mechanism is usually scattering-independent, and therefore, independent of $\sigma_{xx}$ and temperature[5,6].

The established scaling relation is based on the Fermi liquid theory and is expected to apply to weakly interacting systems. The intersection of strong electron correlation and the Berry curvature is still an open territory. On the theory side, due to the interplay between Coulomb repulsion and kinetic degrees of freedom for the electrons, the prediction of band structure properties in strongly correlated materials represents a theoretical challenge. On the experiment side, not many strongly correlated systems have been identified to host large Berry curvature. It is highly demanding to investigate the effect of electron correlation on the Berry curvature-induced physical properties, particularly the scaling relation of the AHE in strongly correlated systems.

Recently Kondo systems have emerged as a promising platform to explore the interaction between strong correlation and Berry curvature associated with Band structure topology. A large anomalous Nernst effect has been observed in a Kondo lattice noncentrosymmetric

[1]Department of Physics, Washington University in St. Louis, St. Louis, MO 63130, USA. [2]Department of Physics, New York University, New York, NY 10003, USA. [3]Beijing National Laboratory for Condensed Matter Physics, Institute of Physics, Chinese Academy of Sciences, Beijing 100190, China. [4]Department of Physics, University of Arizona, Tucson, AZ 85721, USA. [5]University of Nevada, Las Vegas, NV 89154, USA. [6]Ames lab, Ames, IA 50011, USA. [7]Advanced Light Source, Lawrence Berkeley National Laboratory, Berkeley, CA 94720, USA. ✉e-mail: rans@wustl.edu

ferromagnet[16]. A Weyl–Kondo semimetal phase was predicted and observed in non centrosymmetric Kondo semimetal $Ce_3Bi_4Pd_3$[17–21], which exhibits large AHE that is symmetric with respect to applied magnetic fields. Further theoretical studies show that other topological phases, including non-Fermi liquid topological phases, can be driven by the Kondo effect combined with crystalline symmetries[22,23]. Note that in the magnetically ordered systems, Kondo hybridization gives rise to non-symmetry-breaking low-temperature coherence phenomena, which are not precluded by the high transition temperature[24,25].

Here we report the large AHE and breakdown of the scaling relation of AHE in a Kondo lattice ferromagnet, USbTe. The sign change of the AHE upon cooling indicates competing mechanisms from skewing scattering and Berry curvature. The scaling relation between anomalous Hall conductivity and longitudinal conductivity is not valid for a large temperature range. At low temperatures, the scaling relation is recovered which reveals that Berry curvature dominates the AHE. However, Berry curvature contribution has strong temperature dependence and vanishes above the Kondo coherent temperature. This strongly suggests that the Berry curvature in USbTe is hosted by the flat bands at the Fermi level due to Kondo hybridization, consistent with our ARPES measurement and DMFT calculations. Our results demonstrate that the coherence of the flat bands dramatically modifies the scaling relation of AHE, calling for further theoretical investigation of the interaction between strong correlation and Berry curvature-induced physical properties.

## Results

### Kondo lattice ferromagnetism

USbTe crystallizes in a nonsymmorphic crystal structure with space group 129 (P4/nmm). U and Te atoms each form planes with mirror and screw nonsymmorphic symmetries. USbTe exhibits typical behaviors of the ferromagnetic Kondo lattice system. The ferromagnetic ground state develops in USbTe below Curie temperature of $T_c = 125\,K$[26–28], evidenced in resistivity, magnetization, and specific heat measurements (Fig. 1). The high-temperature magnetization follows the Curie-Weiss law with effective magnetic moment $\mu_{eff} = 3.18\,\mu_B$ for $H\|c$, and $3.42\,\mu_B$ for field within $ab$ plane, slightly reduced from the value of a fully degenerate $5f^2$ or $5f^3$ configurations. The anisotropy is readily seen in the paramagnetic region, and becomes more pronounced in the ferromagnetic state. The ordered moment is $\mu_s = 1.93\,\mu_B/U$ for the field along $c$ axis, and only $\mu_s = 0.04\,\mu_B/U$ for the field within $ab$ plane, confirming a dominantly out-of-plane magnetization.

The Sommerfeld coefficient of the electronic contribution to the specific heat is $\gamma_0 = 40\,mJ/mol\text{-}K^2$ (inset to Fig. 1d), indicating that USbTe is a moderately heavy fermion compound. The high-temperature electrical resistivity shows a negative slope, well described by $-c\ln T$, due to the paramagnetic moments in the presence of single-ion Kondo hybridization with the conduction band. Below $T_c$, marked by the kink, resistivity decreases with decreasing temperature. However, the temperature dependence of resistivity can not be described by scattering due to the ferromagnetic magnons, which gives rise to modified exponential behavior, such as $(k_B T/\Delta)^{3/2} Exp(\Delta/_B T)$[29]. Instead, $\rho_{xx}(T)$ shows a pronounced convex shape, not typical for metallic ferromagnets. The previous studies excluded structural change below $T_c$ and attributed the convexity to the formation of Kondo coherent scattering[27,28]. At even lower temperatures, below 15 K, resistivity shows an upturn, which was also observed in the previous study[30]. Likely there remains some incoherent single-ion Kondo scattering together with coherent Kondo scattering at low temperatures.

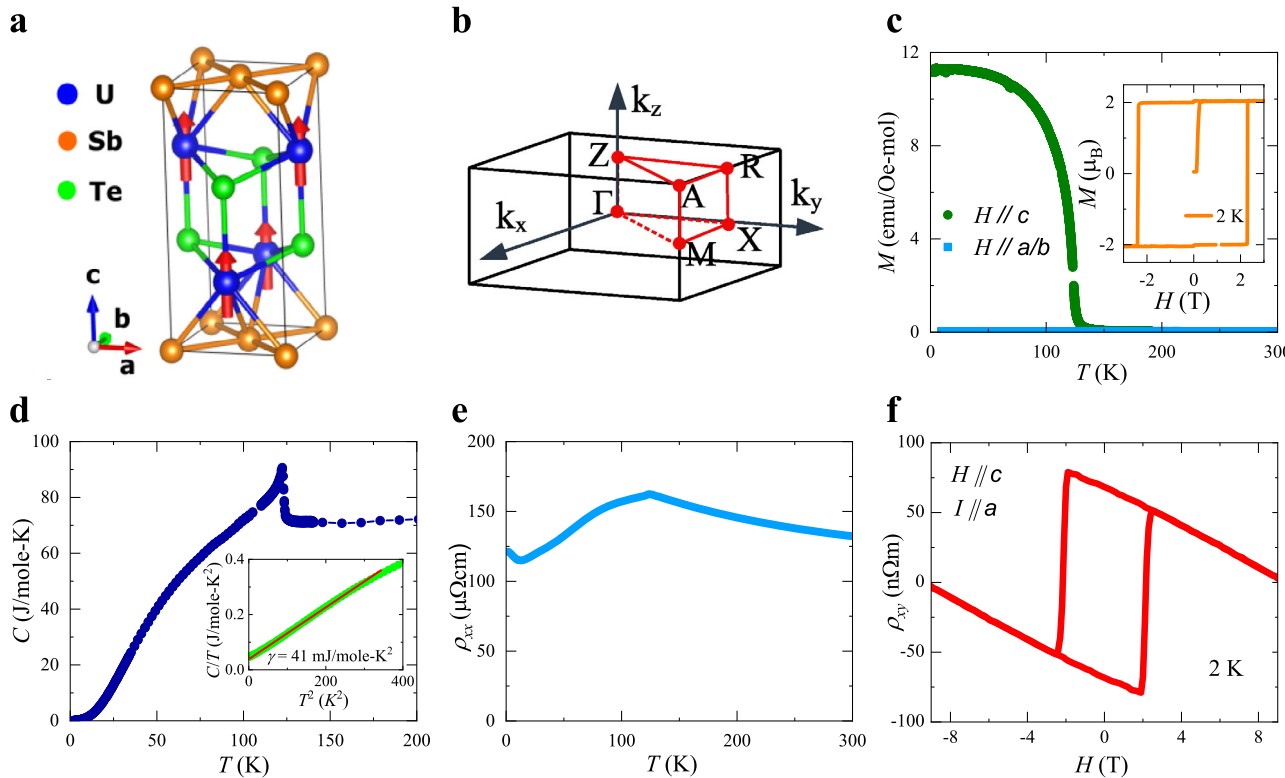

**Fig. 1 | Basic physical properties of USbTe showing ferromagnetic order and Kondo effect. a** Crystal structure of USbTe. **b** First Brillouin zone of USbTe. **c** Magnetization $M$ of USbTe single crystal with a magnetic field of 0.1 T applied along the $c$ axis and $ab$ plane. Inset: Magnetization $M$ of USbTe single crystal as a function of magnetic field at 2 K. **d** Temperature dependence of specific heat $C$ of USbTe single crystal. Inset: $C/T$ as a function of $T^2$ showing the Sommerfeld coefficient as intercept. **e** Temperature dependence of the longitudinal electric resistivity $\rho_{xx}$ in zero-field for USbTe single crystal. Current is applied along $a$ axis. **f** Magnetic field dependence of the Hall resistivity $\rho_{xy}$ of USbTe single crystal at 2 K. Magnetic field is applied along $c$ axis and electric current is along $a$ axis.

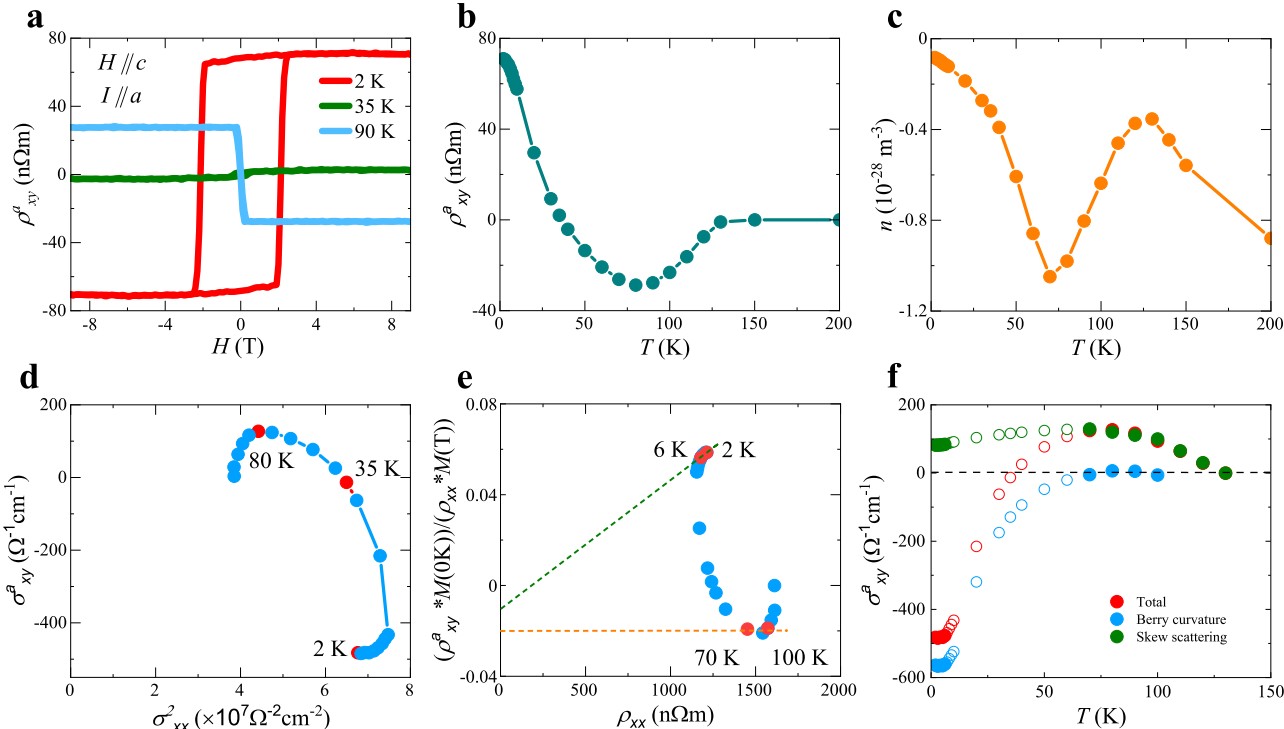

**Fig. 2 | Anomalous Hall effect of USbTe and the scaling analysis. a** Magnetic field dependence of the anomalous Hall resistivity $\rho_{xy}^a$ of USbTe single crystal at three different temperatures, showing a clear sign change of AHE upon cooling, well below $T_c$ of 125 K. Magnetic field is applied along $c$ axis and electric current is along $a$ axis. **b** Temperature dependence of anomalous Hall resistivity $\rho_{xy}^a$ in zero magnetic fields. **c** Carrier density extrapolated from the slope of the Hall resistivity as a function of temperature. Carriers remain to be electrons for the whole temperature range. **d** $\sigma_{xy}^a$ as a function of $\sigma_{xx}^2$, showing that $\sigma_{xy}^a$ is independent of $\sigma_{xx}^2$ below 6 K. **e** $\rho_{xy}^a M(0K)/(M\rho_{xx})$ as a function of $\rho_{xx}$. The slope of the plot represents the intrinsic

Berry curvature contribution to the AHE while the intercept represents the extrinsic skew scattering contribution to the AHE. **f** Estimated contributions to AHE from Berry curvature and skew scattering. In the temperature range of 2–6 and 70–100 K (solid symbols), the separation of intrinsic Berry curvature and extrinsic skew scattering contribution is based on established scaling relation with no additional assumptions. In the temperature range of 6–70 K (circle symbols), the exact temperature dependence of each part is not known and is estimated using the assumption discussed in the main text for simplicity.

## Anomalous Hall effect

Figure 1f shows the Hall resistivity $\rho_{xy}$ of a USbTe single crystal at 2 K, measured with the magnetic field applied along the $c$ axis and the electric current along the $a$ axis. $\rho_{xy}$ exhibits a negative slope indicating a normal Hall effect due to the electron carriers. In addition to the normal Hall effect, a large remnant Hall resistivity at zero-field is observed. The anomalous Hall resistivity $\rho_{xy}^a$ shows rectangular hysteresis loops with very sharp switching, and the coercive field increase with decreasing temperature, resulting in a value of 2 T at 2 K. The most striking feature of the $\rho_{xy}^a$ is the sign change at low temperatures, clearly evident in Fig. 2a, in which we show positive $\rho_{xy}^a$ at 2 K, negative $\rho_{xy}^a$ at 90 K, and almost zero $\rho_{xy}^a$ at 35 K, all well below $T_c$ of 125 K. Figure 2b shows the temperature dependence of the anomalous Hall resistivity $\rho_{xy}^a$, with a broad minimum at 80 K. As temperature decreases, $\rho_{xy}^a$ becomes less negative, passes zero at 35 K, and increases with positive values for even lower temperatures. The sign change in AHE is not due to the change of carrier type, which is related to the normal Hall effect. As a matter of fact, the carriers remain to be electrons in the whole temperature range, evidenced in the negative normal Hall coefficient (Fig. 2c). The sign change in AHE is neither due to the magnetization. In general, AHE could have nonlinear magnetization dependence[2]. However, in our case, magnetization reaches a constant value below 50 K (Fig. 1c) while the sign change of AHE happens at around 35 K.

AHE can be induced by extrinsic mechanisms, i.e., skew scattering and side jump, or intrinsic Berry curvature[1–4]. For a wide range of Kondo lattice systems[31–33], the non-monotonic temperature dependence of Hall resistivity has been observed due to the skew scattering.

At high temperatures, the single-ion Kondo effect gives rise to incoherent scattering of conduction electrons. As the scattering rate does not change much, Hall resistivity will increase upon cooling due to the increase in magnetic susceptibility[31]. This corresponds to the temperature range of 80–125 K in USbTe, with magnetization playing the same role as magnetic susceptibility. At low temperatures, Kondo coherent state forms, and a rapid decrease of Hall resistivity is expected due to the formation of coherent scattering. This corresponds to temperatures below 80 K in USbTe. In a few Kondo lattice systems without magnetic ordering, a significant change of AHE has been observed at low temperatures, which is attributed to the skew scattering in the Kondo coherence regime[34,35]. However, the magnitude of AHE in USbTe is much larger than that in these non-magnetic Kondo lattice systems, and can not be attributed to skew scattering alone.

Scaling analysis has been widely used to distinguish different contributions[1–15]. Skew scattering depends on the scattering rate, leading to a quadratic dependence of anomalous Hall conductivity on the longitudinal conductivity, $\sigma_{xy}^a \sim \sigma_{xx}^2$, while $\sigma_{xy}^a$ originating from the intrinsic scattering-independent mechanism is usually independent of $\sigma_{xx}$[5,6] and temperature. We plot $\sigma_{xy}^a$ as a function of $\sigma_{xx}^2$ in Fig. 2d. Surprisingly, $\sigma_{xy}^a$ vs $\sigma_{xx}^2$ does not seem to follow any scaling relation, nether a linear function of $\sigma_{xx}^2$ nor $\sigma_{xx}$ independent. However, if we focus on the low-temperature region, $\sigma_{xy}^a$ reaches a saturated value below 6 K and is independent of both $\sigma_{xx}$ and temperature, indicating intrinsic Berry curvature contribution.

To further extrapolate the Berry curvature contribution and evaluate the scaling relation, we plot $\rho_{xy}^a/(M\rho_{xx})$ as a function of $\rho_{xx}$,

where $M$ is the magnetization, as shown in Fig. 2e. For ferromagnetic systems, the relation between $\rho_{xy}^a$ and $\rho_{xx}$ should follow $\rho_{xy}^a = a(M)\rho_{xx} + b(M)\rho_{xx}^2$, where the first term corresponds to the skew scattering contribution, and the second term represents the intrinsic contribution[36]. Previous studies indicate that $a(M)$ is proportional to $M$ linearly[36,37]. In general, $b(M)$ could have nonlinear $M$ dependence[2]. However, $M$ dependence of $b(M)$ will not play a role in two temperature ranges, in which magnetization reaches a saturation value or intrinsic contribution vanishes. In these cases, the slope of $\rho_{xy}^a/(M\rho_{xx})$ vs $\rho_{xx}$ plot gives intrinsic Berry curvature contribution, while the intercept gives the skew scattering contribution. Figure 2e shows that $\rho_{xy}^a/(M\rho_{xx})$ is relatively flat between 70 and 100 K, with an unchanged intercept representing skew scattering contribution and no Berry curvature contribution. Below 70 K, the slope starts to deviate from zero, indicating the onset of intrinsic Berry curvature contribution. Below 50 K, the magnetization reaches a constant value. The scaling relation $\rho_{xy}^a = a(M)\rho_{xx} + b(M)\rho_{xx}^2$ reduces to $\rho_{xy}^a = C_1\rho_{xx} + C_2\rho_{xx}^2$ and would predict a linear line for $\rho_{xy}^a/(M\rho_{xx})$ vs $\rho_{xx}$ regardless of the $M$ dependence of $b(M)$. However, this scaling relation is not followed for a large temperature ranges below 50 K as seen in Fig. 2f. The $\rho_{xy}^a = a(M)\rho_{xx} + b(M)\rho_{xx}^2$ relation is only recovered below 6 K, with a finite slope representing intrinsic contribution. The intercept changes from −0.02 at high temperatures to −0.01 at low temperatures, indicating a decrease in the skew scattering contribution. Typically, skew scattering contribution increases as temperature decreases. However, in Kondo lattice systems, Kondo scattering is suppressed at low temperatures due to the formation of the Kondo coherent state, leading to a decrease in skew scattering contribution.

Based on these scaling analyses, we can estimate the Berry curvature and scattering contributions as shown in Fig. 2e. The anomalous Hall conductivity due to the Berry curvature is $560\ \Omega^{-1}$ cm$^{-1}$ at 2 K, $-0.65 \times e^2/hd$ where $d$ is half the lattice constant along the $c$-axis (there are two uranium layers within each unit cell). The large value of anomalous Hall conductivity is comparable to those obtained in weakly correlated Weyl semimetals[7,8,14]. We want to emphasize that the separation of intrinsic and skew scattering contributions to the AHE in the two temperature ranges of interest, 2–6 K and 70–100 K, is based on the established scaling relation with no additional assumptions. Particularly, $M$ dependence of the intrinsic contribution is irrelevant in these two temperature ranges as discussed above. Between 6 and 50 K, $M$ dependence of the intrinsic contribution is still irrelevant since $M$ is constant. However, the scaling relation is violated. There is no established theory predicting how each part of AHE changes with temperature quantitatively in this scenario. We simply assume that anomalous Hall resistivity due to skew scattering changes linearly with temperature from 6 to 70 K. This is only for simplicity. The exact temperature dependence between 6 and 70 K does not change the main finding of the current study and begs for future theoretical investigation.

For completeness, we also need to consider the side jump contribution. The above scaling analysis can not distinguish between intrinsic Berry curvature and side jump contribution. Both mechanisms are independent of the relaxation time of scattering, and have the same scaling analysis. On the other hand, previous studies have shown that the side jump contribution can be estimated using $\gamma_s = ne^2\Delta y/\hbar k_F$[1,38], where $\gamma_s$ is the coefficient for the anomalous Hall resistivity from the side jump $\rho_{xy}^{as} = \gamma_s\rho_{xx}^2$. $\Delta y$ is the amplitude of the side jump in the Hall direction after an impurity scattering event, and is estimated to be $10^{-11}$ m[1,38]. Using the carrier concentration calculated from normal Hall resistivity (supplement material), and the Fermi wavelength from ARPES measurement, we estimate the $\rho_{xy}^{as}$ to be $7\ \Omega^{-1}$ cm$^{-1}$, which yields the anomalous Hall conductivity 50 times smaller than the value we obtained at 2 K. Therefore, the side jump effect is not important in the Kondo coherent regime where Berry curvature dominates the AHE.

The thorough analysis of the anomalous hall data reveals a unique feature of this system: the Berry curvature does not contribute to AHE right below the Curie temperature; It only emerges below Kondo coherent temperature and the magnitude gradually increases at low temperatures until reaching the saturated value. In a large temperature range below Kondo's coherent temperature, AHE does not follow the scaling relation derived from Fermi liquid theory. Our observations strongly indicate that the Berry curvature is from the renormalized bands induced by the Kondo hybridization between $f$ and conduction electrons. To support this idea, we investigated the band structure of USbTe via ARPES measurement and DMFT calculations.

## Band structure revealed by ARPES measurement and DMFT calculations

Band structure mapping with ARPES reveals that the electronic structure is composed of light bands that have significant quasi-2D character (see Methods) and intersect with heavy bands within $E_B < $-30 meV binding energy of the Fermi level (Fig. 3). This class of electronic structure makes $f$-electron coherence an important factor in the emergence of low-temperature physical properties, and the coincidence of $f$-electron states with the Fermi level can occur as a consequence of Kondo physics and/or atomic multiplet correlations[39].

When extrapolated to the Fermi level, the light band features compose three nearly-intersecting Fermi pockets that are traced on the Fermi surface map in Fig. 3a. The electron/hole sign of the light band dispersions is indicated in Fig. 3e. These Fermi surfaces can be interpreted as emerging from a single band with a $[\sqrt{2}, \sqrt{2}]$ reciprocal lattice unit periodicity that matches the Sb sublattice (see yellow dashed lines). This band appears to have a quasi-one dimensional contour when viewed further beneath the Fermi level, and to intersect with a Umklapp-displaced partner at $E_B$ -1.0 eV (see Fig. 3c, d). Band contours at the Fermi surface are difficult to interpret directly from a constant energy map, as photoemission matrix elements are highly inhomogeneous across the Brillouin zone, and the heavy band features have residual intensity at the Fermi level even in regions where they are gapped.

For a closer understanding of the low-energy electronic structure, it is important to trace the energy dependence of the band structure and evaluate more closely how observed band features are gapped from the Fermi level. When the electronic structure is viewed at a high temperature ($T = 135$ K), the light band features appear to have unbroken dispersions that intersect the Fermi level, as traced in black in Fig. 3f, g(top). A flat band feature with limited coherence is found at $E_B$ -20 meV (traced in white). Dispersion of the flat band becomes apparent upon cooling beneath $T < $-50 K and significantly modifies the electronic structure at the Fermi level (see guides to the eye in Fig. 3f, g, bottom). Symmetrizing spectral intensity across the Fermi level at low temperature reveals a gap in all features observed along the $\overline{\Gamma} - \overline{X}$ axis (Fig. 3i, bottom), however, no definitive gap can be resolved in the $\overline{X}$-point pocket (Fig. 3h, bottom). An anomalous dot of intensity near the center of the $\overline{X}$-point pocket is also found to be gapless, and may represent the dispersion minimum of an electron-like pocket residing above the Fermi level. A "gap map" covering the full 2D Brillouin zone is shown in Fig. 3b identifying the binding energy of the shallowest feature observed at all momenta.

We also performed DMFT calculations to compare with ARPES measurement. Figure 4a shows the spectral function of USbTe at $T = 135$ K. The flat U-5$f$ bands appeared in the vicinity of the Fermi level, and were hybridized with the conduction U-6$d$ bands. This gives rise to a kink-like band structure at the Fermi level along the $\overline{X} - \overline{M}$ and $\overline{R} - \overline{A} - \overline{Z}$ high symmetry lines. The distorted conduction bands and flat $f$ bands at the Fermi level are a feature of the Kondo effect[40,41]. The calculated total occupation in the U-5$f$ orbital is 2.22, indicating a strong local magnetic moment of U-5$f$ leads to the formation of the Kondo cloud with spins of conduction electrons.

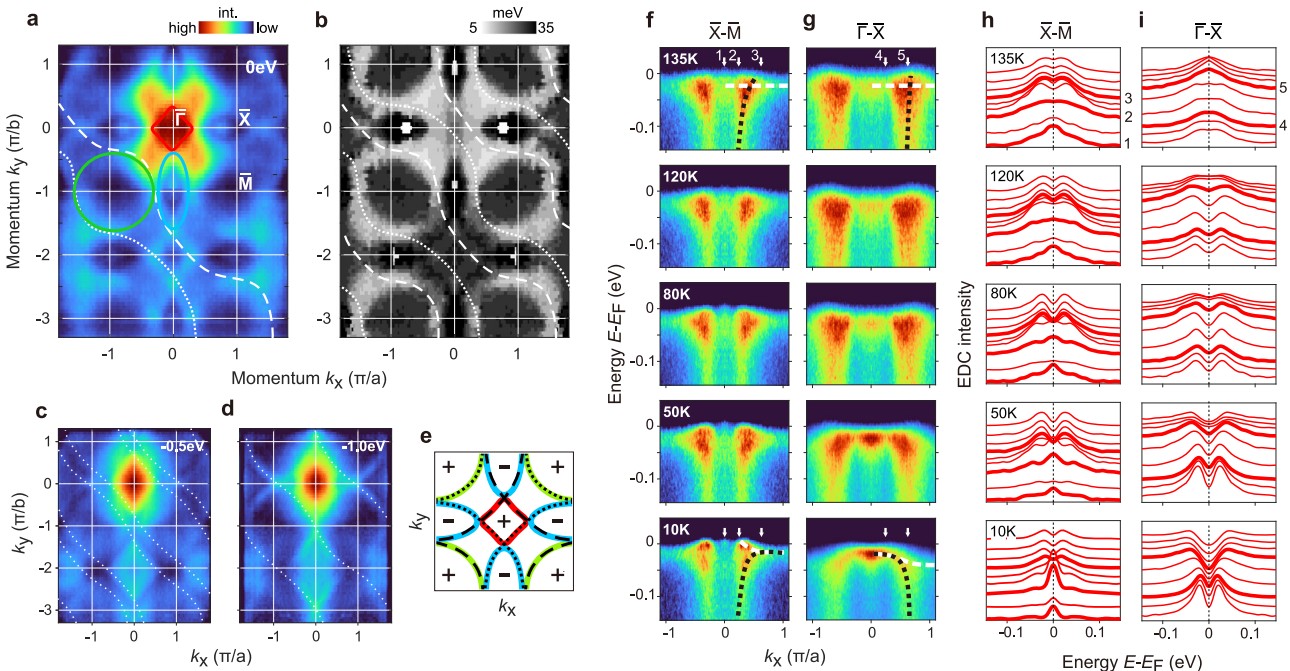

**Fig. 3 | Band structure measurements on USbTe. a** Fermi level ARPES intensity distribution, measured at $T = 20$ K with $h\nu = 98$ eV photons. Guides to the eye are shown for diamond, circle, and oval Fermi contours present at high temperatures ($T > 80$ K). **b** Gap map, showing the binding energy distribution of the closest band feature to the Fermi level at $T = 20$ K (see Methods). **c, d** A light quasi-1D band is traced on constant energy ARPES maps. **e** The high-temperature Fermi surface of the light band traced in **c, d**, with +/- symbols used to indicate hole/electron Fermi pockets. **f, g** Temperature dependence of high-symmetry ARPES measurements. Panels in **g** were obtained at different photon energy ($h\nu = 112$ eV, see Methods). **h, i** Energy dispersion curves from panels **f, g** are symmetrized across the Fermi level to show the presence or absence of gaps. Significant curves are highlighted, corresponding to the white arrows in panels **f, g**.

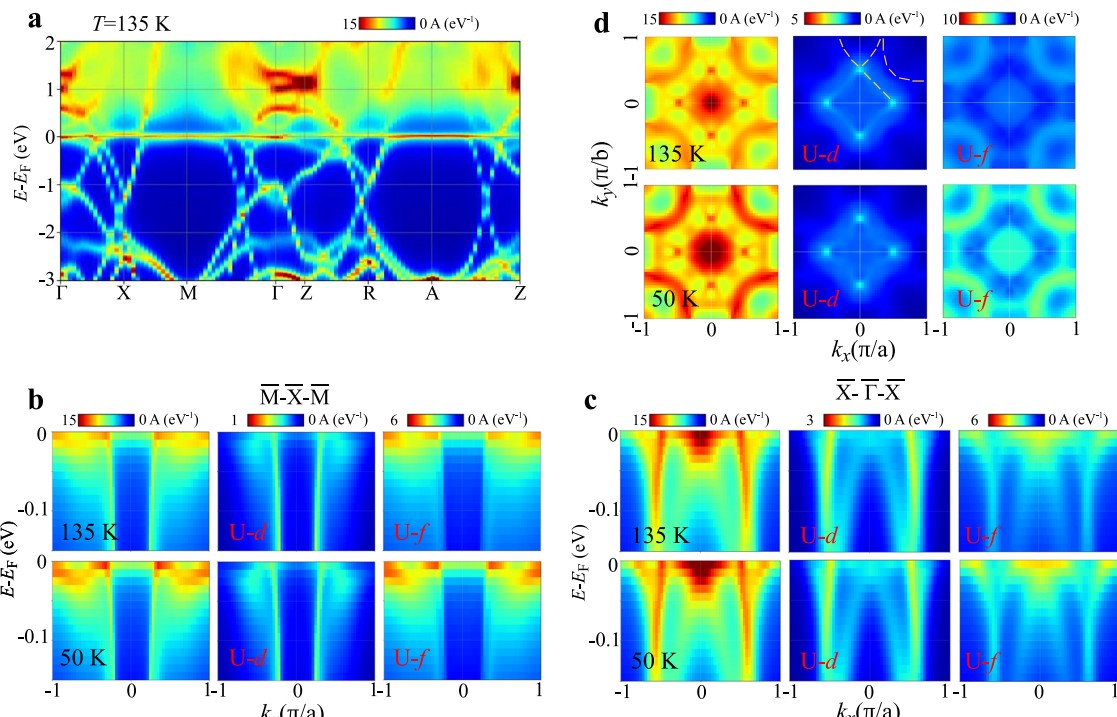

**Fig. 4 | Calculated electronic structure of USbTe. a** Calculated spectral function at 135 K. **b, c** Momentum-resolved spectral function along the **b** $\overline{M} - \overline{X} - \overline{M}$ and **c** $\overline{X} - \overline{\Gamma} - \overline{X}$ at 135 and 50 K. U-6$d$ (U-5$f$) projected spectral function is shown in the middle (right) panel. **d** The calculated Fermi surface at 135 and 50 K. Orbital projected Fermi surfaces are shown. Dash lines are to guide eyes.

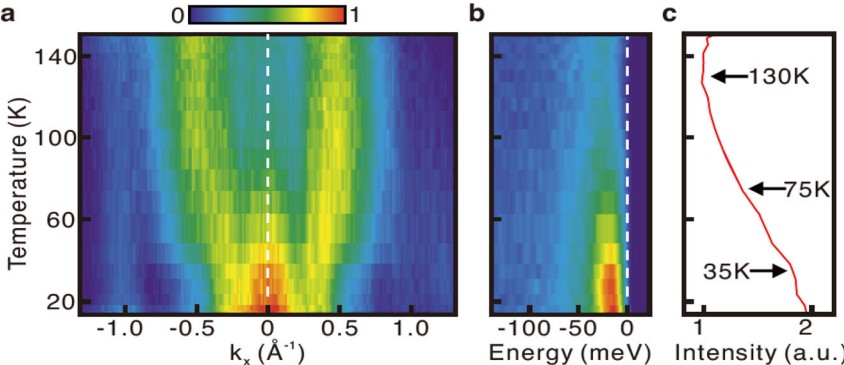

**Fig. 5 | Heavy fermion coherence. a** Temperature dependence of spectral intensity in a 30 meV window at the Fermi level is shown along the $\overline{X} - \overline{\Gamma} - \overline{X}$ momentum axis using $h_\nu = 112$ eV photons. **b** Temperature dependence of ARPES data at the $\overline{\Gamma}$-point reveals the emergence of a low-energy coherence peak. **c** Integrating spectral intensity within ±30 meV of the Fermi level shows a maximal onset slope for the coherence feature at $T \sim 75$K.

We compare the calculated spectral functions at $T = 135$ and $50$ K to ARPES measurements. As shown in Fig. 4b, the U-$6d$ projected spectral functions in $-0.15$ eV $< E - E_f < 0$ along the $\overline{M} - \overline{X} - \overline{M}$ consists of one parabolic band, agreeing with ARPES spectra in the binding energy of $E_B < 0.15$ eV for both temperatures, with the correct mass sign and approximately the same region in momentum space, as shown in Fig. 3f. The more dispersion of the U-$6d$ band at 50 K is apparent and consistent with the ARPES measurements. The dispersion originates from Kondo hybridization with flat $f$ bands, which appear in the U-$5f$ projected spectral function and is stronger at 50 K. Figure 4c shows the U-$6d$ projected spectral functions along the $\overline{X} - \overline{\Gamma} - \overline{X}$. The prominent dispersive spectral feature agrees with ARPES spectra, as shown in Fig. 3g. Some dispersive features present in the $\overline{X} - \overline{\Gamma} - \overline{X}$ simulation are not observed in the ARPES data. However, this is to be expected given the strong incident energy dependence (see Supplementary Fig. 3) and polarization dependence of ARPES matrix elements. The significant contribution of the flat $f$ band in the vicinity of the Fermi level for both temperatures is evident.

Figure 4d shows the calculated Fermi surface in the $k_z = 0$ plane. At $T = 135$ K, the U-$6d$ projected Fermi surface shares similarities with the ARPES-measured Fermi surface as shown in Fig. 3a, e. However, a significant spectral weight of U-$5f$ appears on the calculated Fermi surface. While U-$6d$ states in the Fermi surface are overall hybridized with U-$5f$, prominent U-$5f$ states appear at $\overline{\Gamma}$ and $\overline{M}$ symmetry points only in the U-$5f$ projected Fermi surface. Due to the contribution of dispersive U-$5f$ states to the Fermi surface (see the upper right panel of Fig. 4d), the calculated Fermi surface manifests more spectral weight on the entire Fermi surface than that from the ARPES measurement. In the comparison between calculated Fermi surfaces at 135 and 50 K, the spectral weight of U-$5f$ is stronger at 50 K. This indicates the formation or progress of coherent $f$ bands at the Fermi level signaling renormalization of carriers and coherent Kondo lattice at low temperatures[40,42].

## Discussions

In the weakly correlated ferromagnetic Weyl semimetals, AHE due to the intrinsic mechanism is rather temperature independent[6,7], since it does not dependent on the scattering rate. The scaling relation between $\sigma_{xy}^a$ and $\sigma_{xx}$ follows the theoretical prediction for a large temperature range below $T_c$. This is in sharp contrast to what we observed for USbTe, where AHE due to the Berry curvature has significant temperature dependence and the scaling relation is only recovered at low temperatures.

This extraordinary difference arises from the fact the Berry curvature of USbTe is hosted by the flat bands at the Fermi level formed by Kondo hybridization between U-$5f$ and U-$6d$ electrons. The flat bands at the Fermi level have been clearly shown in our ARPES measurement

and DMFT calculations. These flat bands are subject to the same nonsymmorphic symmetries of the crystal structure of USbTe, which guarantee symmetry-enforced band crossings[22,43]. Together with the spin–orbital coupling and the time-reversal symmetry breaking in the ferromagnetic state, nonsymmorphic symmetries could give rise to Weyl nodes in the flat bands with large Berry curvature. Even without Weyl nodes, a significant spin split of the flat $f$ bands causes Berry curvature as well.

Unlike the light conduction bands hosting Berry curvature in weakly correlated ferromagnetic Weyl semimetals, the coherence of the flat bands in Kondo lattice systems plays an important role in the emergence of low-temperature physical properties. For a system with Kondo temperature $T_K$, the coherence is typically well established for $T < 0.1 \times T_K$, below which the AHE can be described by the Fermi liquid theory and follows the scaling relation. Far Beyond the Kondo temperature, AHE due to the Berry curvature is expected to disappear as a flat band no longer exists. This picture is well consistent with the temperature dependence of the intrinsic AHE of USbTe. The Kondo temperature of USbTe is determined to be around 80 K based on previous thermoelectric measurements. ARPES measurements also show a gradual increase of $f$-electron coherence below 75 K, as seen in Fig. 5. Accordingly, AHE does not have Berry curvature contribution above 80 K, and the scaling relation is recovered below 6 K, $\sim 0.1 \times T_K$.

In the intermediate region with $0.1 \times T_K < T < T_K$, Kondo screening is a crossover versus temperature and the flat bands are broadening with less coherence because of the damping from the thermal effect. The scaling relation between $\sigma_{xy}^a$ and $\sigma_{xx}$ derived from Fermi liquid theory breaks down. A recent theoretical study shows that the damping rate is highly related to the form of the conduction electron self-energy, so is AHE behavior[23]. In the specific model where the Weyl nodes are placed on a two-channel non-Fermi liquid setup, the self-energy has a $\sqrt{T}$ temperature dependence. Subsequently, AHE follows $\sqrt{T}$ at low temperatures[23]. This prediction has not been verified in real material systems yet. Our results on USbTe provide the first experimental evidence that the Kondo coherence dramatically changes the scaling of the AHE, calling for further investigation of the interaction between of strong correlation and Berry curvature-induced physical properties.

## Methods

### Sample synthesis and characterization

Single crystals of USbTe were synthesized by the chemical vapor transport method using iodine as the transport agent. Elements of U, Sb, and Te with atomic ratio 1:0.8:0.8 were sealed in an evacuated quartz tube, together with 1 mg/cm³ iodine. The ampoule was gradually heated up and held in the temperature gradient of 1030/970 °C for 7 days, after which it was furnace cooled to room temperature. The

crystal structure was determined by *x*-ray powder diffraction using a Rigaku *x*-ray diffractometer with Cu-K$_\alpha$ radiation. Electrical transport measurements were performed in a quantum design physical property measurement system (PPMS). Positive and negative magnetic fields were applied in order to antisymmetries the Hall signal. Hall conductivity is calculated using the equation $\sigma_{xy} = -\rho_{xy}/(\rho_{xy}^2 + \rho_{xx}^2)$. Magnetization measurements were performed in a quantum design PPMS with a VSM option. Specific heat measurements were also performed in a quantum design PPMS.

## ARPES measurement

ARPES measurements were performed at the Advanced Light Source MERLIN beamline 4.0.3, with a base pressure similar to $5 \times 10^{-11}$ Torr. Samples were cleaved in situ at $T \sim 20K$. Temperature dependence was obtained by heating to $T = 150$ K and measuring during a subsequent cool-down. Measurements in the main text make use of the uranium *O*-edge resonances at $h_\nu \sim 98$ and 112 eV to enhance sensitivity to uranium *5f* and *6d* electrons.

## LQSGW and DMFT calculations

The electronic structure of USbTe is calculated by employing ab initio linearized quasi-particle self-consistentGW (LQSGW) and dynamical mean field theory (DMFT) method[44–46]. The LQSGW + DMFT is developed based on the full GW + DMFT approach[47–49]. It calculates electronic structure within LQSGW approaches[50,51]. Then, the local strong electron correlation is treated by correcting the local part of GW self-energy within DMFT[52–54]. We use experimental lattice constants of $a = 4.321$ Å and $c = 9.063$ Å[55]. All quantities such as frequency-dependent Coulomb interaction tensor and double-counting energy are calculated explicitly. The local self-energies for U-*6d* and U-*5f* are obtained by solving two different single impurity models utilizing the continuous-time quantum Monte Carlo method. Spin-orbital coupling is included for all calculations. For the LQSGW + DMFT scheme, the code ComDMFT[46] was used. For the LQSGW part of the LQSGW + DMFT scheme, the code FlapwMBPT[51] was used.

## DFT calculations

The spin polarization calculations including the spin−orbital coupling (SOC) correction based on density functional theory (DFT) have been performed through the Vienna ab initio simulation package (VASP) with the Perdew−Burke−Ernzerhof (PBE) generalized gradient approximation (GGA) exchange-correlation potential. The Coulomb interaction of U atoms is considered through the GGA + U method introduced by ref. [56]. For the self-consistent process of electron charge, a proper *k*-mesh of $11 \times 11 \times 6$ is adopted and the energy cutoff of plane wave basis is 350 eV. The single-particle mean-field Hamiltonian is extracted by the *Wannier*90 package from the DFT calculations[57], based on which the anomalous Hall conductivity is calculated through the *WannierTools* package[58] with a dense *k*-mesh of $101 \times 101 \times 101$.

## Data availability

All data generated or analyzed during this study are included in this published article (and its supplementary information files).

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

## Acknowledgements

We are highly indebted to Qimiao Si, Lei Chen, and Chandan Setty for in-depth discussion and continuous theoretical support. We also acknowledge helpful discussions with Xianxin Wu, Heung-Sik Kim, Li Yang, Haonan Wang, Du Li, Linghan Zhu, and Alexander Seidel. L.A.W. acknowledges the support of the National Science Foundation under Grant No. DMR-2105081. This research used resources from the Advanced Light Source, a US DOE Office of Science User Facility under Contract No. DE-AC02-05CH11231. LQSGW and DMFT calculations have been carried out using resources of the National Energy Research Scientific Computing Center (NERSC), supported by the Office of Science of the U.S. Department of Energy under Contract No. DE-AC02-05CH11231 using NERSC award BES-ERCAP0024866. Y.L. and L.K. are supported by the US Department of Energy, Office of Science, Office of Basic Energy Sciences, Materials Sciences and Engineering Division, and Early Career Research Program. Ames Laboratory is operated for the US Department of Energy by Iowa State University under Contract No. DE-AC02-07CH11358. H.W. acknowledges the National Natural Science Foundation of China (Grant No. 11925408, 11921004, and 12188101), the Ministry of Science and Technology of China (Grant No. 2018YFA0305700), the Chinese Academy of Sciences (Grant No. XDB33000000), and the Informatization Plan of Chinese Academy of Sciences (Grant No. CAS-WX2021SF-0102).

## Author contributions

S.R. conceived and directed the project. H.S. synthesized the single crystalline samples. H.S. and C.B. performed the electric transport and magnetization measurements. T.K. performed the specific heat measurements. E.K., S.L., J.D.D., and L.A.W. performed the ARPES measurements. B.K., Q.Z., Y.L., and L.K. performed the DMFT calculations. S.P., H.W., Y.L., and L.K. performed the DFT calculations. H.S., S.R., L.A.W., and B.K. wrote the manuscript with contributions from all authors.

## Competing interests

The authors declare no competing interests.
