## [Peer Review File · Nature Communications]

REVIEWER COMMENTS

Reviewer #1 (Remarks to the Author):

The authors study the anomalous Hall effect in a Kondo lattice ferromagnet USbTe. The anomalous Hall conductivity shows a strong temperature dependence, and the conventional scaling relation of anomalous Hall effect breaks down. Combined with ARPES and DMFT calculations, the authors concluded that the Berry curvature is hosted by the flat bands induced by Kondo hybridization at the Fermi level.

The motivation ("The intersection of strong electron correlation and the Berry curvature is still an open territory") is clear and timely. The large anomalous Hall effect due to the flat bands with the large Berry curvature is interesting and new. I consider that this manuscript potentially deserves the publication in Nat Com. However, I would like to point out that the present analysis of the experimental results is not sufficient.

i) I recommend that the authors discuss the contribution from the side jump mechanism in the main text. In the present version, the side jump is only discussed briefly in SI. The side jump contribution should be important to the conclusion of this paper.

ii) Since the temperature dependence of anomalous Hall resistivity totally deviates from the conventional scaling relation, distinguishing different contributions is in principle difficult. I do not feel that the analysis shown in Fig. 2d-2f is convincing. For example, can the authors explain the temperature dependence of skew scattering contribution shown in Fig. 2f? Intuitively, the skew scattering contribution should increase at lower temperatures; at least, the temperature dependence in Fig. 2f deviates from the textbook trend of the skew scattering contribution. Also, does not Kondo coherent temperature affect the skew scattering?

iii) In SrRuO₃, the anomalous Hall resistivity due to the intrinsic mechanism shows a nonmonotonic temperature dependence and even includes a sign change [Z. Fang et al. Science 302, 92-95 (2003)]; the temperature dependence may be similar to USbTe. Is it possible that the intrinsic mechanism dominates in the whole temperature regime for USbTe?

Reviewer #2 (Remarks to the Author):

This manuscript reports measurements of the anomalous Hall effect (AHE) in a Kondo lattice ferromagnet USbTe.

It is claimed that the AHE is dominated by the intrinsic contribution at low temperatures, namely below the Kondo temperature.

This is presented as evidence that the Berry curvature, contributing to the intrinsic AHE, is hosted by the flat bands, which result

from the Kondo hybridization. I find the results of the paper interesting. However, in my opinion, the data interpretation in the manuscript

is quite ambiguous, since it relies on multiple assumptions of unclear validity. For example, the authors extract the intrinsic contribution

to the AHE from the relation $\rho_{xy} = a(M) \rho_{xx} + b(M) \rho_{xx}^2$, where the first term is the skew scattering contribution while the

second term is the intrinsic contribution. The functions of magnetization $a(M)$ and $b(M)$ are both taken to be linear. I am not sure there is

strong evidence to support this claim. In fact, in many systems exhibiting AHE, e.g. magnetic semiconductors, the dependence of the intrinsic

contribution in the magnetization is in fact strongly nonlinear, and may even be nonmonotonic. I believe this argument in particular needs to be

significantly strengthened before publication can be recommended.

In summary, while I believe this paper does report interesting results, the data analysis and interpretation is too ambiguous in the present version of the manuscript to recommend publication.

Reviewer #3 (Remarks to the Author):

The manuscript by H. Siddiquee reports comprehensive characterization of Kondo lattice ferromagnet USbTe by magnetic, transport, and ARPES measurements. They found large anomalous Hall conductivity originating from the Berry curvature associated with the existence of flat bands near the Fermi level induced by band hybridization. They also found that the anomalous Hall effect shows strong temperature dependence and the scaling relationship breaks down in this compound. Overall, the manuscript is well written, sample characterization is carefully done, and the data quality is high. I have some concerns on the spectral aspects as listed below, which need to be resolved before I recommend the publication of this manuscript in Nature Communications.

1) The first-principles band-structure calculation shown in Fig. S3 shows strong 3D character of USbTe, judged from the sizable band dispersion along the GZ cut. On the other hand, the authors suggest that the observed Fermi surface is essentially quasi-2D (Fig. S2), in contradiction with the band calculation. Can the authors elaborate on the origin of such a difference? Can it be due to electron correlation and/or strong k_z broadening?

2) It is hard to judge from the temperature dependent ARPES intensity in Figs. 3f and 3g that the flat U 5f level and conduction band start to hybridize at low temperature. One may be able to view observed temperature evolution just in terms of the thermal broadening of flat f level and dispersing conduction bands. It would be useful to explicitly exclude this possibility in more convincing way, e.g. by adding more sophisticated analysis of band position by fitting EDCs/MDCs but not by simply showing rough guide lines (which could be somewhat intentional).

3) Although the authors suggest a good agreement between the ARPES data in Fig. 3 and band calculation in Fig. 4, there exist several disagreements such as the gapped and gapless behavior of the spectral feature, magnitude of k_z dispersion, etc. The authors need to be more fair when comparing the experimental data and calculation.

Minor points:

I find several typos on the labeling of figures. e.g.

p.5 , beginning of 2nd paragraph; "Fig. 2d" should be "Fig. 2a".

p. 6, 2nd line of 2nd paragraph; "Fig. 3f" should be "Fig. 2f".

RESPONSE TO REVIEWERS' COMMENTS

Referee 1:

Comment [1]

I recommend that the authors discuss the contribution from the side jump mechanism in the main text. In the present version, the side jump is only discussed briefly in SI. The side jump contribution should be important to the conclusion of this paper.

Our response:

This is a very good suggestion. We moved the discussion of the side jump effect into the main text.

Comment [2]

Since the temperature dependence of anomalous Hall resistivity totally deviates from the conventional scaling relation, distinguishing different contributions is in principle difficult. I do not feel that the analysis shown in Fig. 2d-2f is convincing. For example, can the authors explain the temperature dependence of skew scattering contribution shown in Fig. 2f? Intuitively, the skew scattering contribution should increase at lower temperatures; at least, the temperature dependence in Fig. 2f deviates from the textbook trend of the skew scattering contribution. Also, does not Kondo coherent temperature affect the skew scattering?

Our response:

The referee raised a question regarding the skew scattering contribution. We actually discussed it in the main text (from bottom of page 5 to top of page 6). The reason why it deviates from textbook trend of skew scattering, just as the referee pointed out, is due to the Kondo coherence. Indeed, typically one would expect skew scattering to increase at lower temperatures. However, here the skew scattering comes from Kondo scattering. At lower temperature, basically below Kondo coherent temperature, Kondo scattering becomes coherent, leading to a decrease in skew scattering contribution to AHE. This has been seen in a wide range of Kondo lattice systems, such as UPt₃, UA1₂, CeAl₃, and CeRu₂Si₂, where AHE is due to the Kondo scattering (no intrinsic mechanism). We have cited these compounds in the main text. As this was not explained clearly, we revised the paper to emphasize this point.

The referee also made a general comment regarding our analysis. Although our finding shows that AHE deviates from conventional scaling relation, it does not “totally” deviate. There are two temperature ranges where the scaling relation is followed: 2-6 K and 70-100 K. We used the conventional scaling relation to separate two contributions in this two ranges, and conclude that the conventional scaling relation is violated between 6 and 70 K. In 2-6 and 70-100 K temperature range, our analysis basically follows the standard analysis of AHE data, without other special assumptions. We revised the manuscript to emphasize this.

Comment [3]

In SrRuO₃, the anomalous Hall resistivity due to the intrinsic mechanism shows a nonmonotonic temperature dependence and even includes a sign change [Z. Fang et al. Science 302, 92-95 (2003)]; the temperature dependence may be similar to USbTe. Is it possible that the intrinsic mechanism dominates in the whole temperature regime for USbTe?

Our response:

We would like to thank referee to bring up this system, which provides an excellent reference for comparison. First, we need to clarify two things: **1.** anomalous Hall resistivity of SrRuO₃ indeed shows nonmonotonic temperature dependence. However, nonmonotonic temperature dependence does not necessarily violate the scaling relation. According to the established scaling relation, Hall conductivity ($\sigma_{xy} = -\rho_{xy}/(\rho_{xx}^2 + \rho_{xy}^2)$) due to intrinsic mechanism should be independent of σ_{xx} and temperature. We digitized the data from the paper that referee pointed to, and plot the Hall conductivity as function of temperature and σ_{xx} . It is clear that Hall conductivity is more or less independent of both below 50 K. We further plot Hall conductivity as function of σ_{xx}^2 . It can be seen that at higher temperatures Hall conductivity is close to be linear with σ_{xx}^2 indicating skew scattering mechanism. Please note that the above scaling relation was developed after the Science paper of SrRuO₃ was published. In this Science paper, it was assumed that all of the conductivity is from intrinsic mechanism without further support. In reality, there might always be some skew scattering mechanism, which was not considered in the Science paper. Using this analysis, it is clear that intrinsic mechanism only dominates temperature range below 50 K. To further illustrate this point, we put the data from Co₃Sn₂S₂ (E. Liu et al, Nature Physics 14, 1125–1131, 2018) for comparison. The trend of Hall conductivity of the two system look very similar. Both have intrinsic mechanism dominate the low temperature part, well described by the scaling relation. **2.** Hall resistivity of SrRuO₃ indeed shows a sign change. However, we want to point out that the sign change happens in the vicinity of T_c . Hall resistivity is not zero above T_c , but negative. Then it quickly becomes positive below T_c . Therefore the Hall resistivity around T_c is either due the ferromagnetic fluctuations (or magnetic moment, as no net moment above T_c), or due to the normal Hall effect which was not subtracted from the dataset. Without further information, it is not clear that the sign change is related to the intrinsic mechanism. As a matter of fact, in the paper, the authors mentioned the issues close to T_c in the figure caption of Fig1: “As σ_{xy} should vanish with M at high temperatures, the calculated σ_{xy} is multiplied by the additional M/M_0 factor, which does not affect its behavior except in the vicinity of T_c ”. Therefore it is not clear to what extend their calculations account for the behavior in the vicinity of T_c . Well below T_c , neither their data nor calculations show the sign change.

With these clarified, we want to point out that the behavior in USbTe is very different. Unlike SrRuO₃ and Co₃Sn₂S₂ where the AHE can be scaled either to intrinsic or skew scattering, there is only small temperature range where the established scaling relation works. The sign change of AHE happens well below T_c , again different from SrRuO₃.

Figure 1 a-c data of SrRuO3 from (Z. Fang et al. Science 302, 92-95 (2003)) plotted in different ways. d data of Co3Sn2S2 from (E. Liu et al, Nature Physics 14, 1125–1131, 2018).

Referee 2:

Comment [1]

However, in my opinion, the data interpretation in the manuscript is quite ambiguous, since it relies on multiple assumptions of unclear validity. For example, the authors extract the intrinsic contribution to the AHE from the relation $\rho_{xy} = a(M)\rho_{xx} + b(M)\rho_{xx}^2$, where the first term is the skew scattering contribution while the second term is the intrinsic contribution. The functions of magnetization $a(M)$ and $b(M)$ are both taken to be linear. I am not sure there is strong evidence to support this claim. In fact, in many systems exhibiting AHE, e.g. magnetic semiconductors, the dependence of the intrinsic contribution in the magnetization is in fact strongly nonlinear, and may even be nonmonotonic. I believe this argument in particular needs to be significantly strengthened before publication can be recommended. In summary, while I believe this paper does report interesting results, the data analysis and interpretation is too ambiguous in the present version of the manuscript to recommend publication.

Our response:

The referee raised a very good point. Indeed, we agree that $b(M)$ in general should have more complicated dependence on M . However, we only used this equation to separate skew scattering and intrinsic contribution in two temperature ranges: 2-6 K and 70- 100K. We can safely do this without making any assumptions of M dependence of $a(M)$ and $b(M)$ for the following reasons: **1.** Below 50 K, our M is essentially a constant, therefore in the temperature of 2-50 K, $a(M)$ and $b(M)$ are both constants regardless of the M dependence. The equation simply reduces to $\rho_{xy} = C_1 \rho_{xx} + C_2 \rho_{xx}^2$, from which skew scattering and intrinsic contribution can be extrapolated. Note that our figure 2e has $(\rho_{xy} * M(0K)) / (\rho_{xx} * M(T))$ as y axis. With $M(0K) / M(T) = 1$ below 50 K, it simply reduces to ρ_{xy} / ρ_{xx} with no M dependence. To further illustrate that this analysis is valid regardless of the M dependence as long as M is constant, we apply the same analysis to SrRuO₃, which is a well known example showing nonlinear M dependence of $b(M)$. We digitized the published data [Z. Fang et al. Science 302, 92-95 (2003)] and plot in the same way as our Fig. 2e. It is clear that even though SrRuO₃ has strong nonlinear M dependence, in the low temperature range where M is roughly a constant, it shows a very good linear behavior. This further demonstrates that the analysis we used here is valid regardless of the M dependence as long as M is a constant. In the SrRuO₃ paper, M is actually only approximately constant at low temperatures. In our case, M is really a constant over a wide range of temperatures. We would expect that this analysis should work even better in USbTe below 50 K. The fact that the analysis only works for 2-6 K therefore indicates the breakdown of the conventional scaling relation (not the due to the nonlinear M dependence). **2.** In the temperature range of 70 – 100 K, it is clear from our Fig 2e that $b(M) \rho_{xx}^2$ is zero. Otherwise, there will be a finite slope in this range. The slope may not be proportional to intrinsic contribution if $b(M)$ has nonlinear M dependence. However, since the slope is zero, $b(M)$ is irrelevant anyway. To summarize, in the temperature of 2-50 K and 70- 100 K, our analysis is valid without making any assumptions of $b(M)$.

Figure 2 Data of SrRuO₃ analyzed in the same way as we did for USbTe. Note that in the temperature range where M is constant, the analysis works very well.

We are very glad that referee raised this question. We indeed didn't discuss this well in the previous draft. The above discussion truly strengthens our argument. We revised manuscript accordingly.

In particular, we made it clear that the analysis in the temperature of 2-6 K and 70- 100 K is independent of assumption, while the rest temperature range depends on the assumptions. We change the data points to circle symbols in the rest temperature range. The exact behavior in that range does not change our conclusions and will require future theoretical investigation. We also added the analysis of SrRuO₃ in the SI.

We also want to point out that our analysis of Fig 2d has no assumptions involved. It just plots σ_{xy} as function of σ_{xx}^2 . This figure already illustrates our main conclusion that is the established scaling relation does not work until below 6 K.

Referee 3:

Comment [1]

The first-principles band-structure calculation shown in Fig. S3 shows strong 3D character of USbTe, judged from the sizable band dispersion along the GZ cut. On the other hand, the authors suggest that the observed Fermi surface is essentially quasi-2D (Fig. S2), in contradiction with the band calculation. Can the authors elaborate on the origin of such a difference? Can it be due to electron correlation and/or strong k_z broadening?

Our response:

This is a very good question. The GGA calculation in Fig. S3 includes the Z-G high symmetry axis, from which we can see that $\sim 1/2$ of the bands surrounding the Fermi level have significant z-axis dispersion. The DMFT spectral function in Fig. 4a looks quite different, with a far more 2D character for bands just beneath the Fermi level. Correlations considered by DMFT bring this about in several ways such as: (1) by bringing relatively non-dispersive and quasi-2-dimensional f-states to the Fermi level; (2) by splitting each 5f band into multiple bands that have (3) downward-renormalized creation/annihilation operators and thus reduced dispersion (and reduced spectral weight). All three of these effects cause the observed band structure to be more two-dimensional, however they do not eliminate the expectation that there will be some bands with significant 3D character.

The ARPES measurement in Fig. S2 shows some quasi-2D features, but also shows a number of features that cannot be confidently traced through the Brillouin zone due to strong matrix element effects. These matrix element effects are in part associated with interference with the near-periodic substructures of the unit cell. For example, one expects a beat frequency of $2\pi/d$, where d is the distance between the two uranium sublattices. Broadening along the k_z -axis is also very relevant, but there are no clear-cut indicators that it is a dominant influence in this case, and it can be difficult to assess by eye for a complex band structure.

Since this question is mainly about the GGA+U calculations in the SI, we added this discussion to the SI.

Comment [2]

It is hard to judge from the temperature dependent ARPES intensity in Figs. 3f and 3g that the flat U 5f level and conduction band start to hybridize at low

temperature. One may be able to view observed temperature evolution just in terms of the thermal broadening of flat f level and dispersing conduction bands. It would be useful to explicitly exclude this possibility in more convincing way, e.g. by adding more sophisticated analysis of band position by fitting EDCs/MDCs but not by simply showing rough guide lines (which could be somewhat intentional).

Our response:

This is also a very good question. Indeed, from a theoretical perspective, the hybridization terms in the Green's function are not expected to simply disappear at high temperature -- it is just that they become less relevant to the appearance of the spectral feature and have a far less determinative effect on the DOS distribution at the Fermi level. Our observation is therefore not that hybridization turns on, but rather that the underlying dispersion becomes visually evident.

We do not feel that fitting is practical given the number and nature of the underlying variables, but we have reviewed the text to make sure that precise language is used to discuss this phenomenology. For example, we used language such as "Dispersion of the flat band becomes apparent upon cooling beneath $T < \sim 50\text{K}$ ".

Comment [3]

Although the authors suggest a good agreement between the ARPES data in Fig. 3 and band calculation in Fig. 4, there exist several disagreements such as the gapped and gapless behavior of the spectral feature, magnitude of k_z dispersion, etc. The authors need to be more fair when comparing the experimental data and calculation.

Our response:

Very good point. Following the suggestion, we added more discussions about the points of correspondence and disagreement. We also changed "good agreement" to "share similarities" when there are significant disagreements.

Comment [4]

I find several typos on the labeling of figures. e.g. p.5 , beginning of 2nd paragraph; "Fig. 2d" should be "Fig. 2a". p. 6, 2nd line of 2nd paragraph; "Fig. 3f" should be "Fig. 2f".

Our response:

Thanks for pointing out these typos. We have fixed them.

REVIEWERS' COMMENTS

Reviewer #1 (Remarks to the Author):

I confirmed that the authors have carefully revised the manuscript in response to my previous comments. This manuscript is now ready for publication.

Reviewer #2 (Remarks to the Author):

The authors have addressed my comments in a satisfactory manner and I believe the paper may now be published in its present form.

Reviewer #3 (Remarks to the Author):

I have read through the revised manuscript and authors' responses to my comments. In my opinion, the authors have satisfactorily addressed to all of my concerns regarding the difference between ARPES and DFT results and the temperature-dependent ARPES intensity. The manuscript has been appropriately revised. Now the manuscript would be suitable for the publication in Nature Communications.